# Beyond Correctness: Rewarding Faithful Reasoning in Retrieval-Augmented Generation

**Zhichao Xu**[1*]                                          *xzhichao@amazon.com*

**Zongyu Wu**[2*†]                                          *zongyuwu@psu.edu*

**Yun Zhou**[1]

**Aosong Feng**[1]

**Kang Zhou**[1]

**Sangmin Woo**[1]

**Kiran Ramnath**[1]

**Yijun Tian**[1]

**Xuan Qi**[1]

**Weikang Qiu**[3†]

**Lin Lee Cheong**[1]

**Haibo Ding**[1]                                          *hbding@amazon.com*

[1] *AWS AI Fundamental Research*       [2] *The Pennsylvania State University*       [3] *Yale University*

**Reviewed on OpenReview:** *https://openreview.net/forum?id=mZ0gGlXelF*

## Abstract

Inspired by the success of reinforcement learning (RL) in Large Language Model (LLM) training for domains like math and code, recent work has begun training LLMs to dynamically plan, query, and reason with search engines as tools — a paradigm increasingly referred to as *agentic search*. Although these methods achieve performance improvement across popular short-form QA benchmarks, many prioritize final answer correctness while overlooking the quality of intermediate reasoning steps, which may lead to *chain-of-thought unfaithfulness*. In this paper, we first introduce a comprehensive evaluation framework for agentic search, covering three distinct faithfulness metrics: *Think-Search faithfulness*, *Information-Think faithfulness*, and *Think-Answer faithfulness*. Our evaluations reveal that canonical agentic search systems trained through Reinforcement Learning from Verifiable Reward (RLVR) using episode-level outcome-based reward — including SEARCH-R1 and RESEARCH — have significant room for improvement on these faithfulness dimensions. To foster faithful reasoning in agentic search, we introduce VERITAS (Verifying Entailed Reasoning through Intermediate Traceability in Agentic Search), a novel framework that integrates fine-grained turn-level faithfulness rewards into the reinforcement learning process. Our experiments show that models trained with VERITAS not only significantly improve reasoning faithfulness, but also achieve better task performance compared to baselines trained against episode-level outcome-based reward.

## 1 Introduction

Large Language Models (LLMs) have achieved remarkable success across diverse domains (Brown et al., 2020; Georgiev et al., 2024; Grattafiori et al., 2024), yet they continue to face persistent challenges such as hallucinations (Li et al., 2023; Maynez et al., 2020; Huang et al., 2025) and outdated knowledge (Mousavi

---

*Both authors contributed equally to this work.

†Work done during an internship at Amazon Web Services.

et al., 2024). While Retrieval-Augmented Generation (RAG, Lewis et al., 2020; Zhao et al., 2024) mitigates these issues by grounding outputs in external knowledge, conventional retrieve-then-generate pipelines often fail on complex reasoning queries that require multi-turn interaction and evidence synthesis (Gao et al., 2023; Trivedi et al., 2023; Yao et al., 2023). This has motivated a shift toward *agentic search* (Singh et al., 2025; Liang et al., 2025) — a paradigm in which LLMs are trained via reinforcement learning to dynamically plan, issue queries to a retriever, and reason over returned evidence as part of an interactive trajectory (Jin et al., 2025b; Li et al., 2025; Chen et al., 2025; Song et al., 2025).

Despite impressive benchmark results, existing agentic search methods still face a critical limitation: they optimize for final-answer correctness (Jin et al., 2025b; Chen et al., 2025; Li et al., 2025) while overlooking the faithfulness of intermediate reasoning steps (Baker et al., 2025; Bao et al., 2025). As recent work reveals (Lanham et al., 2023; Bentham et al., 2024), outcome-based training often produces reasoning traces that misalign with the final answer, a phenomenon termed as *chain-of-thought unfaithfulness*. Maintaining Chain-of-Thought (CoT) faithfulness (Lanham et al., 2023; Baker et al., 2025) and contextual faithfulness (Malaviya et al., 2025) are critical for reasoning models, particularly for RL-based search agents, as the intermediate CoT steps provide users with a clear reasoning pathway to understand how a complex query is decomposed into atomic queries used for retrieving evidence and leads to the final answer that addresses the advanced information need (Marchionini, 2006). However, it remains unclear how reasoning faithfulness in the context of agentic search should be formulated and evaluated.

To systematically address this ambiguity, we first formalize and quantify faithfulness in agentic search. We propose a novel evaluation framework centered on three key dimensions of reasoning traceability: **Think-Search** faithfulness (*are search queries justified by the agent's thoughts?*), **Information-Think** faithfulness (*do the agent's thoughts consider the retrieved information?*), and **Think-Answer** faithfulness (*is the final answer grounded in the agent's thoughts?*). Applying this framework to popular agentic search models such as SEARCH-R1 (Jin et al., 2025b) and RESEARCH (Chen et al., 2025), reveals a critical gap: as shown in Figure 1, the gain in the final-answer accuracy does not automatically guarantee the faithfulness of intermediate reasoning steps.

Motivated by recent work showing that augmenting episode-level outcome rewards with feedback on intermediate reasoning can improve RL training (Cobbe et al., 2021; Lightman et al., 2024; Zhang et al., 2025), we hypothesize that optimizing only outcome-based rewards while overlooking the quality of intermediate reasoning steps may hinder model performance. However, applying faithfulness as a training signal in agentic search is non-trivial: a single trajectory contains heterogeneous output types (thoughts, queries, retrieved evidence, answers), each of which requires a different notion of correctness. We therefore introduce **VERITAS** (**V**erifying **E**ntailed **R**easoning through **I**ntermediate **T**raceability in **A**gentic **S**earch), a framework designed to train verifiably faithful search agents. The name, derived from the Latin word for "truth" (*vēritās*), reflects our objective to ensure the agent's reasoning is true to the evidence it gathers. VERITAS operationalizes this ideology by incorporating our proposed faithfulness metrics directly into the RL training loop as fine-grained, structured faithfulness rewards.[1] Our trained model, VERITAS-R1, not only improves information-think faithfulness by around 14% and think-answer faithfulness by around 7.7% over its comparable SEARCH-R1 baseline, but also increases task accuracy.

In summary, our contributions are:

- We propose a formal definition and a corresponding evaluation framework for faithfulness in agentic search, centered on three novel metrics: Information-Think, Think-Answer, and Think-Search Faithfulness.

- Through this framework, we conduct a comprehensive analysis of leading RL-based search agents, revealing a gap between their task performance and the reasoning faithfulness.

---

[1] We use the terms *structured faithfulness rewards* or *checklist-style process rewards* throughout the paper, in contrast to the per-step process reward models of Lightman et al. (2024) and Cobbe et al. (2021), which score every reasoning step. VERITAS instead defines a small fixed set of faithfulness criteria tied to specific transitions in the agentic search trajectory (think→search, information→think, think→answer), which is more closely related to recent rubric- and checklist-based evaluation (Ribeiro et al., 2020; Arora et al., 2025; Deshpande et al., 2025) and rubric-as-reward RL (Gunjal et al., 2026; Viswanathan et al., 2025; Shao et al., 2025). All three lines of work share the goal of contrasting with episode-level outcome rewards, but differ in granularity and structure. We discuss this connection in detail in § 6.

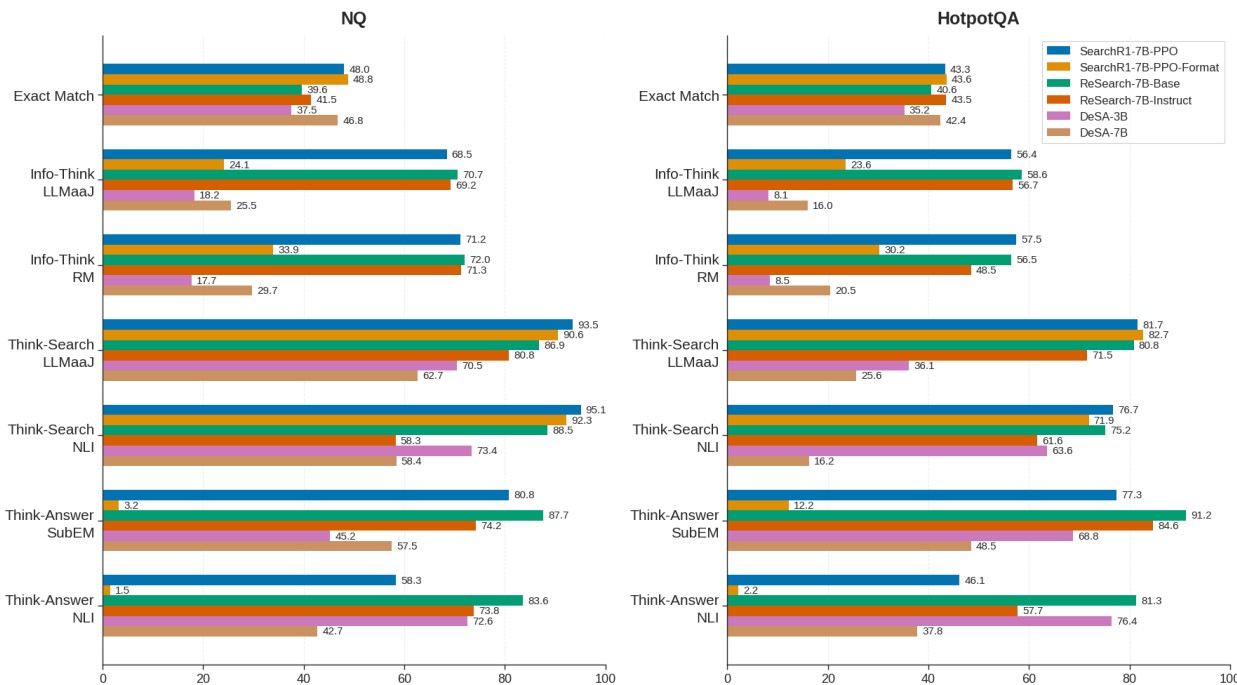

Figure 1: Evaluation of SEARCH-R1, RESEARCH and DESA on task performance (Exact Match) and faithfulness dimensions. Left: results on NQ dataset. Right: results on HotpotQA dataset. We can see models trained with pure outcome-based reward, their good task performance does not automatically guarantee fully faithful reasoning chains — notably the Information-Think faithfulness dimension — which may be suboptimal for the practical deployment of such systems.

- We introduce VERITAS, a novel training framework that integrates fine-grained faithfulness metrics as structured, checklist-style rewards into the RL loop. Our experiments show that the proposed method effectively improves the model's faithfulness while maintaining comparable task accuracy across seven downstream QA benchmarks.

## 2 Related Work

**Faithfulness in reasoning and contextualization.** Although explicit reasoning can enhance LLM performance, the presence of intermediate steps does not necessarily reveal the true causal mechanisms behind model outputs (Turpin et al., 2023; Lanham et al., 2023; Balasubramanian et al., 2025). A key question is whether these steps genuinely support the final answer or merely serve as plausible post-hoc explanations. This concern has motivated a growing body of research on *Chain-of-Thought (CoT) faithfulness* (Atanasova et al., 2023; Paul et al., 2024; Balasubramanian et al., 2025; Chua & Evans, 2025; Parcalabescu & Frank, 2024). Counterfactual approaches (Atanasova et al., 2023; Chen et al., 2024) analyze how perturbing reasoning steps affects model predictions, while adversarial methods (Lanham et al., 2023; Matton et al., 2025) intervene on the CoT or model states to probe causal dependencies (Balasubramanian et al., 2025). Another popular direction focuses on bias articulation and explanation reliability (Turpin et al., 2023; Balasubramanian et al., 2025; Arcuschin et al., 2025).

Beyond reasoning traces, LLMs often exhibit *contextual unfaithfulness* — producing outputs that contradict or are unsupported by given contexts (Huang et al., 2025; Bi et al., 2025; Ye et al., 2023). This phenomenon limits their reliability in applications requiring factual grounding. Prior studies (Dziri et al., 2022b;a; Ming et al., 2025; Malaviya et al., 2025) have evaluated whether model responses are properly entailed by reference passages, emphasizing accurate synthesis without introducing unsupported claims. Together, these works

reveal the importance of verifying both reasoning and contextual grounding, yet they primarily focus on static text generation rather than dynamic, search-driven reasoning.

**Retrieval-Augmented Generation.** Retrieval-Augmented Generation (RAG, Lewis et al., 2020; Gao et al., 2023; Jiang et al., 2023; Trivedi et al., 2023; Fan et al., 2024; Chan et al., 2024; Li et al., 2025; Jiang et al., 2025; Chang et al., 2025; Xu et al., 2025c) enables LLMs to access external knowledge for more factual and up-to-date responses. RAG techniques span diverse directions (Gao et al., 2023), including query rewriting (Ma et al., 2023; Zheng et al., 2024a; Xu et al., 2025d), iterative retrieval (Li et al., 2025), and supervised fine-tuning for tool use (Schick et al., 2023; Asai et al., 2024; Patil et al., 2024). Despite their effectiveness on QA and knowledge-intensive tasks (Lewis et al., 2020; Xu et al., 2025a), many RAG systems depend heavily on prompt engineering or are difficult to scale (Sun et al., 2025). More critically, they are typically evaluated on final-answer accuracy, overlooking whether the intermediate reasoning steps remain faithful to the retrieved evidence.

**Agentic Search with Reinforcement Learning.** Reinforcement Learning (RL, Watkins & Dayan, 1992; Kaelbling et al., 1996; Sutton et al., 1998) has been widely adopted to align model behavior with long-term objectives. Early work explored RL from human feedback (Stiennon et al., 2020; Ouyang et al., 2022; Gheshlaghi Azar et al., 2024; Kaufmann et al., 2025), often using Proximal Policy Optimization (PPO, Schulman et al., 2017). Recent methods such as REINFORCE Leave-One-Out (RLOO, Kool et al., 2019; Ahmadian et al., 2024), Group Relative Policy Optimization (GRPO, Shao et al., 2024; Yu et al., 2026) improve stability and sample efficiency by removing the need for a separate critic, while off-policy preference optimization methods (Rafailov et al., 2023; Meng et al., 2024) are considered more scalable but lag behind in terms of performance (Lanchantin et al., 2025; Lin et al., 2025).

Chain-of-Thought prompting (Wei et al., 2022; Kojima et al., 2022) decomposes complex tasks into interpretable steps (Geva et al., 2021; Chu et al., 2024) to achieve better performance. ReAct (Yao et al., 2023) asks the LLM-based agent to reason before taking actions. Based on these technical approaches, recent works have explored training reasoning-focused LLMs (Jaech et al., 2024). For instance, DeepSeek-R1 (Guo et al., 2025) employs GRPO to generate long reasoning chains that solve challenging queries. Inspired by this success, several works have applied RL to CoT-based reasoning, training LLMs to interact with search tools (Chen et al., 2025; Jin et al., 2025b; Song et al., 2025). While outcome-based approaches (Jin et al., 2025b; Li et al., 2025) optimize for final-answer correctness, others design intermediate rewards to encourage more deliberate search or reasoning (Jin et al., 2025a; Zheng et al., 2025; Shi et al., 2025b; Zhao et al., 2025b). For example, AutoRefine (Shi et al., 2025b) introduces a retrieval-specific reward that encourages models to utilize evidence effectively — a signal conceptually related to faithfulness.

However, no prior work has comprehensively evaluated or formalized the notion of faithfulness in RL-based agentic search. To fill this gap, we propose an evaluation framework consisting of three distinct faithfulness dimensions for agentic search use case; and further proposes a training-based approach that integrates these faithfulness dimensions as fine-grained, structured rewards at key trajectory transitions.

# 3 A Framework for Evaluating Faithfulness in Agentic Search

As noted in § 1, a critical limitation of current RL-based search agents is that the policy trained with pure outcome-based reward may elicit unfaithful reasoning. To systematically diagnose this issue, we need to establish a formal evaluation. We first brief the essential background, then introduce our proposed evaluation framework.

## 3.1 Background: Agentic Search Trajectory

In agentic search, LLMs dynamically use search engines as tools. We select SEARCH-R1 (Jin et al., 2025b) as a canonical example (algorithm details in Appendix C). The interaction process produces a structured trajectory containing text blocks wrapped with four types of tags (Jin et al., 2025b):

- `<think></think>`: The agent's internal reasoning, where it analyzes information and plans its next step.
- `<search></search>`: The search query generated by the agent to send to the retriever.

- `<information></information>`: The retrieved information from the search tool.
- `<answer></answer>`: The agent's final answer to the initial query.

Since the RL-based search agent is purely trained to optimize the outcome-based reward (e.g., the correctness of the final answer), it overlooks the integrity of this intermediate trajectory. To address this gap, we develop a framework to systematically measure the faithfulness of the intermediate reasoning steps.

## 3.2 Faithfulness Definitions

To systematically analyze the quality of the reasoning process in agentic search, we introduce a structured framework based on three distinct, turn-by-turn notions of faithfulness. These definitions provide a theoretical grounding for our subsequent evaluation. Let a trajectory be a sequence of alternating thoughts, search queries, and retrieved information, culminating in a final answer. We define faithfulness at three critical junctures within this process. Formally, we define the following three types of faithfulness in the context of agentic search:

**Think-Search faithfulness.** This dimension assesses the logical connection between the agent's internal reasoning process and its subsequent search query. A search query is considered faithful if it is a direct and necessary consequence of an information gap explicitly identified or implicitly raised in the immediately preceding thought process. The core principle is that the agent should "search what it thinks it needs", ensuring that its information-gathering actions are deliberate and well-motivated by its reasoning state.

**Information-Think faithfulness.** This dimension evaluates whether the agent's reasoning properly uses the evidence it has just acquired. The thinking block following an information block is considered faithful if the reasoning it contains is a valid synthesis, summary, or logical deduction based on the newly retrieved information. This form of faithfulness is crucial for preventing the model from ignoring evidence, thereby ensuring the integrity of each step in the reasoning chain.

**Think-Answer faithfulness.** This dimension measures whether the final answer is supported by the cumulative knowledge gathered throughout the entire trajectory. The final answer is deemed faithful if the final claim it makes is entailed by the content within the *latest preceding thinking block*. This formulation ensures that the agent does not introduce new, unverified information at the final step and that its conclusion is attributed to the complete, self-contained search and reasoning trajectory.

## 3.3 Evaluation Metrics

To operationalize the faithfulness dimensions defined in § 3.2, we design evaluation metrics that align with the semantic role of each transition in the agentic search trajectory. We combine lightweight automatic checks with LLM-based judgment, reflecting the different levels of semantic subtlety required across faithfulness types. Metrics are organized following the temporal order of the trajectory.

- **Think–Search Faithfulness.** Think–Search faithfulness assesses whether a search action is logically motivated by the immediately preceding reasoning. We first formulate this dimension as a textual entailment problem, treating the `<think></think>` block as the *premise* and the subsequent `<search></search>` query as the *hypothesis*. Using a pretrained NLI model, `google/t5_xxl_true_nli_mixture` (Honovich et al., 2022), we label each (`<think></think>`, `<search></search>`) pair as Entailment or No entailment. Because strict entailment may undercount faithful searches that are implicitly motivated or differently abstracted, we additionally employ an LLM-as-a-Judge (LLMaaJ) evaluation. The judge LLM assesses whether the search query is a reasonable and well-motivated response to the agent's reasoning, flagging irrelevant or disconnected queries. Unless otherwise specified, we use `Claude Sonnet-4.5` as the judge, with the prompt shown in Appendix Table 6. We report both NLI-based and LLMaaJ-based scores and analyze their agreement.

- **Information–Think Faithfulness.** Information–Think faithfulness evaluates whether retrieved evidence is meaningfully incorporated into subsequent reasoning. As this requires assessing synthesis and selective use of information, we adopt an LLM-as-a-Judge approach. The judge is presented with

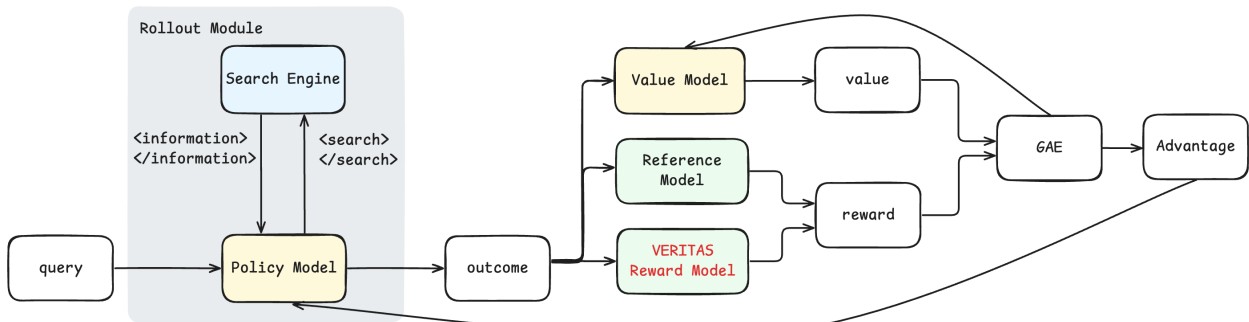

Figure 2: The pipeline of VERITAS-R1 in the RL framework with PPO RL algorithm. Applying process supervision (VERITAS Reward Model) improves the faithfulness of policy model's intermediate reasoning steps. Outcome in this figure refers to the rollout trajectory used for subsequent reward calculation and advantage estimation.

(`<information></information>`, `<think></think>`) pairs and determines whether the reasoning is grounded in, consistent with, and responsive to the retrieved content, flagging ignored, contradicted, or unsupported claims. The output is a binary judgment. The prompt template for judge model is provided in Appendix Table 7.

- **Think–Answer Faithfulness.** Think–Answer faithfulness measures whether the final answer is supported by the immediately preceding reasoning. We operationalize this dimension using two automatic metrics. First, we apply an NLI-based entailment check, treating the final `<think></think>` block as the *premise* and the `<answer></answer>` block as the *hypothesis*. Second, for short-form QA settings, we use a substring exact-match (Sub-EM) metric that verifies whether key entities or factual claims in the answer explicitly appear in the preceding reasoning. We examine the agreement between these metrics in § 3.4 and discuss their respective trade-offs.

**Generalizability.** Currently, the proposed metrics are instantiated for SEARCH-R1, but the underlying framework is broadly applicable to any reasoning-centric system that produces *structured trajectories*, such as RESEARCH (Chen et al., 2025) and DESA (Wang et al., 2025b). Because it operates over the abstract *think–search–information–answer* schema, the evaluation methodology remains agnostic to model architecture and domain, which enables consistent assessment of turn-level faithfulness across diverse multi-turn retrieval and reasoning agents. This generalizability also allows the framework to serve as a foundation for developing structured, checklist-style training objectives, as demonstrated in the next section through the design of VERITAS.

### 3.4 Faithfulness Evaluation Results

To have a basic understanding of the faithfulness of existing agentic search models, we evaluate three RL-based RAG models: SEARCH-R1 (Jin et al., 2025b), RESEARCH (Chen et al., 2025) and DESA (Wang et al., 2025b) trained against outcome-based rewards. We use their official checkpoints to ensure reproducibility. For Think-Search faithfulness and Think-Answer faithfulness, we also calculate the agreement rate and Fleiss's $\kappa$ as meta-evaluation of the two evaluation metrics. From Figure 1, we make these key observations.

- **The Information-Think Faithfulness of evaluated models are far from perfect**. For example, SEARCH-R1-7B-Base PPO has an Information-Think faithfulness score of 0.564 on the complex multi-hop HotpotQA dataset, while the best checkpoint we evaluated — RESEARCH-7B-Base achieves 0.586, suggesting that *the policy models trained against pure outcome-based reward are not fully faithful to the retrieved information during the thinking process.*
- **Task performance does not directly translate to high faithfulness scores**. For example, DESA-7B achieves competitive performance on both NQ and HotpotQA datasets, but it consistently underperforms SEARCH-R1 and RESEARCH in almost all three faithfulness dimensions.

Table 1: Meta-evaluation results on NQ and HotpotQA datasets. For SEARCH-R1, we use the checkpoints using the Base model. For Think-Search and Think-Answer faithfulness, the Fleiss $\kappa$ tends to be lower due to the imbalanced class distribution.

| Dataset | Model | EM | Info-Think | | | | Think-Search | | | | Think-Ans | | | |
|---------|-------|----|-----------|------|-------|-------|-------------|------|-------|-------|-----------|------|--------|-------|
| | | | LLMaaJ | RM | Agree. | Kappa | LLMaaJ | NLI | Agree. | Kappa | SubEM | NLI | Agree. | Kappa |
| NQ | SearchR1-7B-PPO | 0.480 | 0.685 | 0.712 | 0.880 | 0.689 | 0.935 | 0.951 | 0.947 | 0.058 | 0.808 | 0.583 | 0.756 | 0.456 |
| | SearchR1-7B-PPO-Format | 0.488 | 0.241 | 0.339 | 0.732 | 0.384 | 0.906 | 0.923 | 0.913 | 0.029 | 0.032 | 0.015 | 0.975 | 0.458 |
| | ReSearch-7B-Base | 0.396 | 0.707 | 0.720 | 0.878 | 0.645 | 0.869 | 0.885 | 0.933 | 0.241 | 0.877 | 0.836 | 0.873 | 0.484 |
| | ReSearch-7B-Instruct | 0.415 | 0.692 | 0.713 | 0.875 | 0.638 | 0.808 | 0.583 | 0.746 | 0.075 | 0.742 | 0.739 | 0.897 | 0.731 |
| | DeSA-3B | 0.375 | 0.182 | 0.177 | 0.962 | 0.881 | 0.705 | 0.734 | 0.910 | 0.773 | 0.452 | 0.726 | 0.464 | -0.027 |
| | DeSA-7B | 0.468 | 0.255 | 0.297 | 0.867 | 0.673 | 0.627 | 0.584 | 0.878 | 0.745 | 0.575 | 0.427 | 0.719 | 0.450 |
| HotpotQA | SearchR1-7B-PPO | 0.433 | 0.564 | 0.575 | 0.847 | 0.666 | 0.817 | 0.767 | 0.831 | 0.133 | 0.774 | 0.461 | 0.680 | 0.385 |
| | SearchR1-7B-PPO-Format | 0.436 | 0.236 | 0.302 | 0.817 | 0.575 | 0.827 | 0.719 | 0.825 | 0.181 | 0.122 | 0.022 | 0.898 | 0.262 |
| | ReSearch-7B-Base | 0.406 | 0.586 | 0.565 | 0.833 | 0.607 | 0.808 | 0.752 | 0.872 | 0.311 | 0.912 | 0.813 | 0.874 | 0.479 |
| | ReSearch-7B-Instruct | 0.435 | 0.567 | 0.485 | 0.808 | 0.593 | 0.715 | 0.616 | 0.873 | 0.719 | 0.846 | 0.577 | 0.713 | 0.051 |
| | DeSA-3B | 0.352 | 0.081 | 0.085 | 0.974 | 0.846 | 0.361 | 0.636 | 0.488 | 0.048 | 0.688 | 0.764 | 0.838 | 0.586 |
| | DeSA-7B | 0.424 | 0.160 | 0.205 | 0.910 | 0.711 | 0.256 | 0.162 | 0.796 | 0.402 | 0.485 | 0.378 | 0.792 | 0.583 |

- **All evaluated checkpoints have high Think-Search faithfulness**. For instance, SEARCH-R1 has a Think-Search score of 0.935 on NQ dataset when using LLMaaJ. The results suggest that their search query aligns well with their reasoning process.

We also note that SEARCH-R1-7B-Base-PPO-Format (Jin et al., 2025a) shows poor Information-Think faithfulness. A closer manual examination of its traces suggested that for the `<think> </think>` block after retrieved information, the policy often directly transitions to the next search query without reasoning over the documents, e.g., "`<information>`...`</information><think>`I need to find out the name of the first person who got the Nobel Prize in Physics.`</think>`". The policy also tends to continue to search until reaching the maximum number of allowed search turns, then being forced to generate an answer, leading to poor Think-Answer faithfulness, e.g., "`<think>`I need to find out who got the first Nobel Prize in Physics.`</think><answer>` Wilhelm Röntgen `</answer>`". These observations collectively reveal that *a modified reward shaping by adding a format reward may improve model performance, but in return potentially breaks the model's reasoning consistency.*

## 3.5 Meta-evaluation

As we evaluate RL-based RAG models, we are also conducting meta evaluation of our evaluation metrics. We report detailed meta evaluation results in Table 1. While using a powerful LLM-as-a-Judge provides high-quality faithfulness labels, its cost and latency make it impractical for scalable evaluation. For Information-Think faithfulness, we propose to use a smaller reward model by distilling from the LLM judge, which we leave details to § 4. For Think-Search faithfulness, we observed high agreement rate between the LLM judge and the NLI model, thus we propose to use the NLI model as the automatic evaluation. For Think-Answer faithfulness, we notice a moderate inconsistency between Sub-EM and NLI metric. A closer manual evaluation suggests that as the RAG datasets primarily focus on short-form answers, the NLI model trained on symmetric (`Premise`, `Hypothesis`) pairs often predict "No entailment" for coherent think-answer pairs, leading to high False Positive rate. Therefore, we adopt Sub-EM as the automatic metric for Think-Answer faithfulness evaluation.

Our findings in this section highlight that, despite their strong answer accuracy, the three RL-based RAG models we examined still exhibit notable weaknesses in reasoning faithfulness — particularly in how retrieved evidence is integrated into the thought process. This observation underscores a fundamental limitation of existing agentic search models trained solely with outcome-based rewards: They fail to incentivize faithful reasoning traces to arrive at final answers.

# 4   VERITAS: Training Faithful Search Agents with Structured Faithfulness Rewards

Our analysis in § 3 revealed a critical gap: RL agents trained with purely outcome-based rewards often fail to produce faithful reasoning, even when they arrive at the correct final answer. This finding highlights the need for direct supervision of intermediate reasoning quality. To fill this gap, we introduce VERITAS, a framework designed to train search agents that are not only correct but also verifiably faithful. An overview of the pipeline with PPO RL algorithm is shown in Figure 2. The core principle of VERITAS is to enrich the RL reward signal with fine-grained feedback at a small, fixed set of structurally meaningful transitions in the agentic search trajectory — a checklist-style alternative to per-step process reward models. This is achieved through two key components: a multi-faceted reward function that balances task accuracy with reasoning faithfulness, and a practical, distilled reward model to efficiently supervise the RL training.

**Reward design.**   The foundation of VERITAS is a reward function that combines the conventional outcome-based reward with our proposed structured faithfulness rewards. Denote the outcome-based reward as $\mathcal{R}_{EM}$, measured by the exact match between the predicted answer and the groundtruth as in SEARCH-R1. For the two key faithfulness dimensions we optimize during training, we define their corresponding rewards as $\mathcal{R}_{info\text{-}think}$ and $\mathcal{R}_{think\text{-}answer}$. The final reward function is a weighted sum:

$$\mathcal{R} = w_{EM} \cdot \mathcal{R}_{EM} + w_{info\text{-}think} \cdot \mathcal{R}_{info\text{-}think} + w_{think\text{-}search} \cdot \mathcal{R}_{think\text{-}search} + w_{think\text{-}answer} \cdot \mathcal{R}_{think\text{-}answer}, \quad (1)$$

where $w$ are hyperparameters representing the weight of each corresponding reward component. This reward formulation explicitly optimizes the agent to balance task accuracy (EM) and the faithfulness of its reasoning process. We ground our implementation of VERITAS on the SEARCH-R1 framework.

Practically, we opted to set $w_{think\text{-}search} = 0$, i.e., skipping the Think-Search term. The reason of this choice is three-fold: (i) quantitatively, we observed in § 3 that all checkpoints we evaluated are already showing high Think-Search faithfulness; (ii) in our manual inspections of the SEARCH-R1 trajectories, we found that the model has developed a consistent pattern of "first reason about its information need, then write the search query" (case studies in Appendix H), suggesting that training with EM reward already equipped the policy model with robust think-search faithfulness; (iii) training search agents is time-consuming and computationally expensive. Removing this term eliminates an extra hyperparameter, thereby streamlining the training process and enhancing the efficiency.

We note that VERITAS can be considered a conceptual framework that incorporates faithfulness as a structured reward signal to improve the search agent. The exact instantiation of each individual reward can be implemented differently. In this work, we use a model-based approach for information-think faithfulness and substring exact-match for think-answer faithfulness, and leave more exploration of other reward implementations — such as rich semantic matching signals for think-answer faithfulness — to future work.

**Practical implementation via a distilled reward model.**   The cost and latency of LLMaaJ approach make it impractical for scalable on-policy RL training. To tackle this challenge, a core component of the VERITAS framework is a smaller, distilled reward model (RM) trained to replicate the judgments of the larger LLMaaJ. We focus exclusively on information-think faithfulness.

To train the RM, we first collect a large-scale dataset of SEARCH-R1 trajectories. We subsample 27,000 samples from a combination of the NQ (Kwiatkowski et al., 2019) and HotpotQA (Yang et al., 2018) training splits. We prompt Claude-3.7-Sonnet to label these instances for Information-Think faithfulness (see Appendix for prompt templates). We then split the 27K instances into 24K for training and 3K for evaluation. Using LoRA (Hu et al., 2022), we fine-tune a Qwen2.5-14B-Instruct model, which has high consistency (0.899) and a strong Cohen's $\kappa$ coefficient (0.797) with the Claude-3.7-Sonnet labels. To validate our RM, an author of this paper manually annotated 100 examples from the test set. Both Claude-3.7-Sonnet and our trained RM show high consistency with human judgments (0.850 and 0.910, respectively, with corresponding Cohen's $\kappa$ of 0.699 and 0.819; see Table 11). Given its strong performance and efficiency, we use this fine-tuned RM for all subsequent RL training and evaluation.

Further, note that the datasets for Info-Think RM are labeled by Claude-3.7-Sonnet rather than Claude-4.5-Sonnet which is the judge model in § 3.4. There are two main reasons: (i) The API of Claude-3.7-Sonnet becomes deprecated when we are extending the faithfulness evaluation to more agentic search models. Re-

Table 2: Performance of different methods on selected QA datasets. $^{\dagger}$ denotes in-domain datasets and $^{*}$ denotes out-of-domain dataset. All the baseline results are from Jin et al. (2025b). VERITAS-R1 is SEARCH-R1 trained with our VERITAS framework. ♣ indicates statistically significant compared to Search-R1-7B-Base-PPO, while ♠ indicates significant compared to Search-R1-7B-Base-PPO w/ Format.

| Methods | General QA | | | Multi-Hop QA | | | | Avg. |
|---|---|---|---|---|---|---|---|---|
| | NQ$^{\dagger}$ | TriviaQA$^{*}$ | PopQA$^{*}$ | HotpotQA$^{\dagger}$ | 2wiki$^{*}$ | MuSiQue$^{*}$ | Bamboogle$^{*}$ | |
| Direct Inference | 0.134 | 0.408 | 0.140 | 0.183 | 0.250 | 0.031 | 0.120 | 0.181 |
| CoT | 0.048 | 0.185 | 0.054 | 0.092 | 0.111 | 0.022 | 0.232 | 0.106 |
| IRCoT | 0.224 | 0.478 | 0.301 | 0.133 | 0.149 | 0.072 | 0.224 | 0.239 |
| Search-o1 | 0.151 | 0.443 | 0.131 | 0.187 | 0.176 | 0.058 | 0.296 | 0.206 |
| RAG | 0.349 | 0.585 | 0.392 | 0.299 | 0.235 | 0.058 | 0.208 | 0.304 |
| SFT | 0.318 | 0.354 | 0.121 | 0.217 | 0.259 | 0.066 | 0.112 | 0.207 |
| R1-base | 0.297 | 0.539 | 0.202 | 0.242 | 0.273 | 0.083 | 0.296 | 0.276 |
| R1-instruct | 0.270 | 0.537 | 0.199 | 0.237 | 0.292 | 0.072 | 0.293 | 0.271 |
| Rejection Sampling | 0.360 | 0.592 | 0.380 | 0.331 | 0.296 | 0.123 | 0.355 | 0.348 |
| DeSA-3B-Instruct-GRPO | 0.375 | 0.575 | 0.397 | 0.352 | 0.363 | 0.134 | 0.347 | 0.363 |
| DeSA-7B-Instrct-GRPO | 0.468 | 0.631 | 0.440 | 0.424 | 0.374 | 0.197 | 0.395 | 0.418 |
| ReSearch-7B-Base | 0.396 | 0.606 | 0.442 | 0.406 | 0.447 | 0.217 | 0.432 | 0.422 |
| ReSearch-7B-Instruct | 0.415 | 0.640 | 0.450 | 0.435 | **0.476** | **0.223** | 0.424 | 0.438 |
| Search-R1-7B-Base-PPO w/ Format | **0.488** | 0.644 | **0.469** | 0.436 | 0.412 | 0.187 | 0.403 | 0.434 |
| Search-R1-7B-Base-PPO | 0.480 | 0.638 | 0.457 | 0.433 | 0.382 | 0.196 | 0.432 | 0.431 |
| *VERITAS-R1* | | | | | | | | |
| EM+Info-Think | 0.486 | 0.650 | 0.463 | 0.445 | 0.423 | 0.206 | 0.456 | **0.447**♣♠ |
| EM+Think-Ans | 0.482 | **0.658** | 0.464 | 0.445 | 0.420 | 0.189 | 0.416 | 0.439 |
| EM+Info-Think+Think-Ans | 0.484 | 0.645 | 0.466 | **0.446** | 0.419 | 0.192 | **0.464** | 0.445♣ |

labeling the large datasets using Claude-4.5-Sonnet and retraining the RM on the newly labeled data can be costly. (ii) Our evaluation results in Table 10 show that Claude-3.7-Sonnet and Claude-4.5-Sonnet have high consistency ratio and $\kappa$ coefficient, meaning that the two models are highly consistent. Therefore, we still use the datasets labeled by Claude-3.7-Sonnet for RM's training and evaluation. Further training details and evaluation results, including the human evaluation analysis are in Appendix F.

**Training curriculum.** In preliminary RL experiments, we found that directly applying the faithfulness rewards $\mathcal{R}_{info\text{-}think}$ and $\mathcal{R}_{think\text{-}ans}$ substantially improved faithfulness metrics but slightly reduced task performance compared to SEARCH-R1. We hypothesize that introducing these rewards too early restricts the policy model's exploration of diverse reasoning strategies, causing it to over-optimize for faithfulness — a form of reward hacking. To mitigate this, we adopt a curriculum learning-inspired strategy: from step 1 to $T_1$, we use only outcome-based rewards by setting $w_{info\text{-}think}$ and $w_{think\text{-}ans}$ in Equation (1) to 0; from $T_1 + 1$ to $T_2$, we linearly warm up these weights (note that this reward warmup is separate from the linear learning rate schedule); and from $T_2$ onward, we apply the full rewards. Practically, $T_1$ is set to the end of the first training epoch, and the warmup lasts 0.5 epoch.

To summarize, the VERITAS framework provides a practical methodology for structured supervision of intermediate reasoning by combining a multi-faceted reward function with an efficient, distilled reward model. This enables direct optimization of reasoning faithfulness during RL training without sacrificing performance or scalability.

# 5   Experiments

We conduct a series of experiments to validate our central hypothesis: that incorporating structured faithfulness rewards via the VERITAS framework can improve both the reliability of an agent's reasoning and its final task performance. We focus on the following key research questions:

**RQ1:** Does training with VERITAS lead to improvement in reasoning faithfulness?

**RQ2:** How do these structured rewards affect final task accuracy compared to purely outcome-based RL training?

## 5.1 Experiment Setup

**Models and datasets.** To ensure a fair comparison, we build our implementation directly upon the SEARCH-R1 framework. We use the Qwen2.5-7B-Base model (Yang et al., 2024) as our policy model and PPO (Schulman et al., 2017) as the RL algorithm, mirroring the best-performing configuration of the original SEARCH-R1. Evaluation spans seven diverse QA benchmarks: NQ (Kwiatkowski et al., 2019), TriviaQA (Joshi et al., 2017), PopQA (Mallen et al., 2023), HotpotQA (Yang et al., 2018), 2Wiki (Ho et al., 2020), MuSiQue (Trivedi et al., 2022), and Bamboogle (Press et al., 2023). These datasets cover a range of complexities from general to multi-hop QA, allowing for a comprehensive assessment.

**Baselines.** We compare our model, which we refer to as **VERITAS-R1** (i.e., SEARCH-R1 trained with the VERITAS framework), against a comprehensive set of baselines. Our primary comparison is against the original SEARCH-R1 (Jin et al., 2025b), which uses outcome-only RL. We also include other baselines, detailed in Appendix E.

**Metrics.** To answer our research questions, we evaluate all models on: (i) *Task Performance*, measured by Exact Match (EM), and (ii) *Reasoning Faithfulness*, using our proposed Information-Think and Think-Answer faithfulness metrics (§ 3.3). For task performance, we report dataset-level paired Wilcoxon signed-rank tests across the seven benchmarks ($n = 7$) at the $p = 0.05$ level, following standard practice for comparing learned models over multiple datasets (Demšar, 2006; Dror et al., 2018). We adopt this dataset-level paired protocol rather than aggregating across training seeds because (i) RL training of search agents is computationally intensive — each run requires tens of high-memory GPUs over several days — making large seed sweeps infeasible, and (ii) hypothesis tests over a small number of seeds are known to have very low statistical power and produce unreliable conclusions (Henderson et al., 2018). As a stability check, we additionally re-ran the Info-Think variant of VERITAS-R1 with three independent seeds and observed average EM of $\{0.442, 0.446, 0.445\}$ (std 0.0021), consistent with the result reported in Table 2.

**Implementation details.** Our reward model is a LoRA-finetuned Qwen2.5-14B-Instruct (Yang et al., 2024), trained as described in § 4 using LLaMA-Factory (Zheng et al., 2024b). For the VERITAS-R1 policy model, we tune the reward weights and find an optimal balance with $w_{EM} = 1.0$, $w_{info\text{-}think} = 0.05$, and $w_{think\text{-}ans} = 0.02$ (see Figure 4 for an ablation). Other training parameters, such as learning rate (1e-6) and batch size (256), as well as the retriever (E5-base-v2 (Wang et al., 2022)) and corpus (2018 Wikipedia dump), are kept identical to the original SEARCH-R1 setup for a controlled comparison. All experiments were conducted on 32x NVIDIA A100 GPUs.

## 5.2 Results and Analysis

Our results demonstrate that VERITAS successfully improves both faithfulness and task accuracy, confirming our central hypothesis. We find that explicitly rewarding the reasoning process creates a synergistic effect, leading to more robust and effective agents.

**VERITAS improves reasoning faithfulness (RQ1).** As shown in Figure 3, training with VERITAS rewards leads to gains in faithfulness. For example, incorporating the $\mathcal{R}_{info\text{-}think}$ reward can largely improves Information-Think faithfulness across all dataset categories. For example, on general QA datasets, average Information-Think faithfulness increases from 0.734 for the baseline SEARCH-R1 to 0.853 for VERITAS-R1. Interestingly, VERITAS-R1 further enhances SEARCH-R1's Think-Search Faithfulness on both general QA and Multi-Hop QA datasets, even though we do not adopt think-search faithfulness as the training signal. This directly validates that our structured faithfulness rewards successfully steer the agent towards generating more grounded and verifiable reasoning steps. Gains in think–answer faithfulness are clear on multi-hop datasets. However, the improvement of think–answer faithfulness is not stable on general QA datasets.

**Improved faithfulness translates to higher accuracy (RQ2).** Crucially, the improvement in reasoning quality translates to better final performance. As detailed in Table 2, VERITAS-R1 consistently outperforms

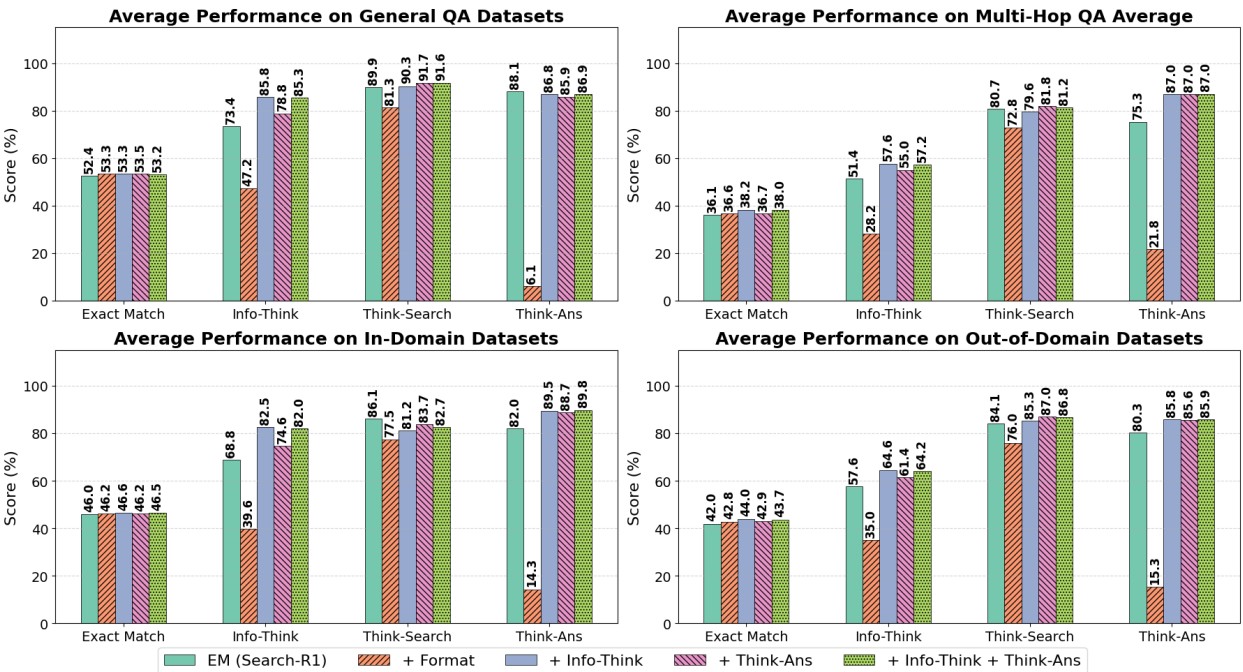

Figure 3: Faithfulness evaluation results comparing two variants of Search-R1 and our methods.

Table 3: Model Performance of the proposed Faithful Search-R1. (EM results are from our machines and may differ slightly from Table 2 due to vLLM randomness).

| Method | NQ | TriviaQA | PopQA | HotpotQA | 2wiki | MuSiQue | Bamboogle | Average |
|---|---|---|---|---|---|---|---|---|
| *Exact Match (EM)* | | | | | | | | |
| Search-R1 (Baseline) | 0.482 | 0.641 | 0.450 | 0.437 | 0.381 | 0.201 | 0.425 | 0.431 |
| + Format (Baseline) | 0.494 | 0.643 | 0.462 | 0.430 | 0.413 | 0.196 | 0.425 | 0.436 |
| VERITAS Info-Think | 0.486 | 0.650 | 0.463 | 0.445 | 0.423 | 0.206 | 0.456 | 0.447 |
| VERITAS Think-Ans | 0.482 | 0.658 | 0.464 | 0.442 | 0.420 | 0.189 | 0.416 | 0.439 |
| VERITAS Info-Think+Think-Ans | 0.484 | 0.645 | 0.466 | 0.446 | 0.419 | 0.192 | 0.464 | 0.445 |
| *Info-Think Faithfulness* | | | | | | | | |
| Search-R1 (Baseline) | 0.762 | 0.745 | 0.695 | 0.614 | 0.349 | 0.442 | 0.651 | 0.608 |
| + Format (Baseline) | 0.429 | 0.534 | 0.452 | 0.364 | 0.190 | 0.256 | 0.317 | 0.363 |
| VERITAS Info-Think | 0.933 | 0.821 | 0.819 | 0.718 | 0.354 | 0.490 | 0.744 | 0.697 |
| VERITAS Think-Ans | 0.795 | 0.778 | 0.792 | 0.698 | 0.332 | 0.452 | 0.717 | 0.652 |
| VERITAS Info-Think + Think-Ans | 0.929 | 0.811 | 0.820 | 0.712 | 0.356 | 0.489 | 0.732 | 0.693 |
| *Think-Search Faithfulness* | | | | | | | | |
| Search-R1 (Baseline) | 0.950 | 0.867 | 0.880 | 0.771 | 0.744 | 0.846 | 0.867 | 0.846 |
| + Format (Baseline) | 0.857 | 0.774 | 0.808 | 0.693 | 0.739 | 0.676 | 0.802 | 0.764 |
| VERITAS Info-Think | 0.855 | 0.919 | 0.934 | 0.769 | 0.745 | 0.848 | 0.821 | 0.842 |
| VERITAS Think-Ans | 0.882 | 0.933 | 0.935 | 0.792 | 0.769 | 0.855 | 0.857 | 0.860 |
| VERITAS Info-Think+Think-Ans | 0.879 | 0.934 | 0.934 | 0.775 | 0.774 | 0.851 | 0.848 | 0.856 |
| *Think-Ans Faithfulness* | | | | | | | | |
| Search-R1 (Baseline) | 0.836 | 0.898 | 0.909 | 0.803 | 0.644 | 0.670 | 0.895 | 0.808 |
| + Format (Baseline) | 0.052 | 0.064 | 0.066 | 0.235 | 0.327 | 0.195 | 0.113 | 0.150 |
| VERITAS Info-Think | 0.892 | 0.844 | 0.868 | 0.899 | 0.872 | 0.819 | 0.889 | 0.869 |
| VERITAS Think-Ans | 0.879 | 0.855 | 0.842 | 0.895 | 0.869 | 0.838 | 0.877 | 0.865 |
| VERITAS Info-Think+Think-Ans | 0.893 | 0.838 | 0.877 | 0.903 | 0.867 | 0.823 | 0.888 | 0.870 |

the strong SEARCH-R1 baseline across the majority of datasets. On the challenging multi-hop datasets, VERITAS-R1 improves the average EM score from 0.361 to 0.380. The average EM score on general QA

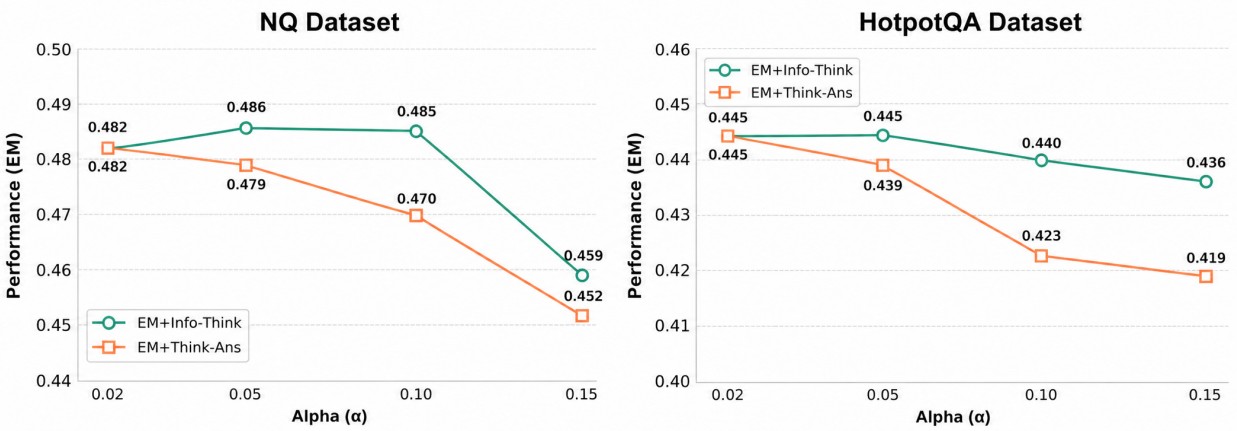

Figure 4: Hyperparameter sensitivity study.

Table 4: Effects of the training curriculum to task performance. We use the VERITAS Info-Think variant to show the tradeoff between task performance and Info-Think faithfulness between two training strategies.

| Method | NQ | TriviaQA | PopQA | HotpotQA | 2wiki | MuSiQue | Bamboogle | Average |
|---|---|---|---|---|---|---|---|---|
| *Exact Match (EM)* | | | | | | | | |
| w/o curriculum | 0.471 | 0.641 | 0.480 | 0.437 | 0.429 | 0.174 | 0.400 | 0.433 |
| w/ curriculum | 0.486 | 0.650 | 0.463 | 0.445 | 0.423 | 0.206 | 0.456 | 0.447 |
| *Info-Think Faithfulness* | | | | | | | | |
| w/o curriculum | 0.956 | 0.832 | 0.826 | 0.786 | 0.595 | 0.619 | 0.749 | 0.766 |
| w/ curriculum | 0.933 | 0.821 | 0.819 | 0.718 | 0.354 | 0.490 | 0.744 | 0.697 |

datasets is also increased from 0.524 to 0.532. This demonstrates that encouraging the model to "think" more faithfully is not a constraint but a facilitator for finding the correct answer, effectively reducing reward hacking and promoting more robust problem-solving strategies. The detailed performance on all seven datasets can be found at Table 3.

## 5.3 Ablation Studies

**Analysis of reward components.** Our analysis also reveals nuances in how different faithfulness rewards affect behavior. While the Information-Think reward generally provides a positive signal for faithfulness and EM, the Think-Answer reward has a more complex effect. As seen in Figure 3, although adding $\mathcal{R}_{think-answer}$ improves its corresponding metric on a macro average (0.865 vs 0.808 averaged over 7 datasets), it leads to a slight drop on TriviaQA and PopQA datasets. This suggests that ensuring thoughts are grounded in evidence ($\mathcal{R}_{info-think}$) is a more effective and stable mechanism for improving overall search model quality than enforcing a strict logical entailment to the final answer during training.

**Hyperparameter sensitivity.** We experiment with different hyperparameters $w_{info-think}, w_{think-ans}$ and report results in Figure 4. We notice that increasing both hyperparameters to greater than 0.05 leads to performance degradation, as the policy model overly focus on the faithfulness reward. Therefore, we opted to use $w_{info-think} = 0.05$, $w_{think-ans} = 0.02$ in our final combined reward (EM+Info-Think+Think-Ans).

**Effects of the training curriculum.** We show an ablation study of the training curriculum in Table 4. We use a simplified setting to only compare two reward signals — EM and Info-Think faithfulness — and their corresponding evaluation results. We notice that directly adding the faithfulness reward at the start of the training makes the policy model learn to hack the faithfulness reward, leading the lower EM results at the end of the training. The training dynamics (Figure 5) also suggest that directly adding the faithfulness reward at the start of the training limits the policy model's exploration.

### 5.4 Discussions

**Limitation and improvement of model-based evaluations.** Although using LLMs to evaluate results has been a common practice, it has been reported in the literature that they still have several limitations (Li et al., 2024), such as position bias (Shi et al., 2025a), token/label bias (Jiang et al., 2024; Xu et al., 2024a) and overconfidence (Khan et al., 2024). Recent works have proposed corresponding mitigations, such as calibration (Wang et al., 2024; Lee et al., 2025) or aggregating predictions from multiple LLMs (Verga et al., 2024; Zhao et al., 2025a).

In this work, we opted for a uni-dimensional scalar evaluation for each individual faithfulness dimension. Recent works have also explored multi-dimensional, structured evaluation criteria, commonly referred to as "*checklists*" or "*rubrics*" (Ribeiro et al., 2020; Arora et al., 2025; Deshpande et al., 2025), and further incorporate this fine-grained signals into RL training (Gunjal et al., 2026; Viswanathan et al., 2025; Shao et al., 2025, *inter alia*). Due to the limited bandwidth, we leave a more comprehensive investigation of faithfulness evaluation and corresponding RL training to the future work.

We also note that our evaluation pipeline is explicitly designed to mitigate circularity between the training and evaluation signals. The held-out judge used throughout § 3 (Claude-4.5-Sonnet) differs from the labeling source of our distilled reward model (Claude-3.7-Sonnet, see § 4), so the faithfulness numbers reported in § 3 are decoupled from the training reward. We additionally validate the distilled reward model against human annotations (Table 11), where it achieves 0.910 consistency and Cohen's $\kappa$=0.819 over 100 samples — higher than either Claude-3.7-Sonnet or Claude-4.5-Sonnet on the same set — suggesting it serves as a reliable proxy rather than introducing arbitrary bias.

## 6 Conclusion and Future Works

In this work, we addressed the issue of unfaithful reasoning in RL-based agentic search, where models optimized for final answers often produce untrustworthy intermediate steps. We introduced a formal evaluation framework with three faithfulness metrics and proposed VERITAS, a training paradigm that integrates these metrics as structured, checklist-style rewards. Our central finding is that rewarding the reasoning process does not trade off with performance; instead, it creates a positive synergy across diverse QA benchmarks, leading to agents that are not only more faithful but also achieve higher task accuracy. This work underscores the value of structured supervision over intermediate reasoning for developing more reliable and transparent AI agents.

More broadly, VERITAS can be viewed as decomposing a holistic notion of trajectory quality — faithfulness — into a small set of structured, rubric-like sub-criteria (think–search, information–think, think–answer), each instantiated as an independent reward signal. This connects naturally to the emerging line of rubric- and checklist-based evaluation (Ribeiro et al., 2020; Arora et al., 2025; Deshpande et al., 2025) and rubric-as-reward RL (Gunjal et al., 2026; Viswanathan et al., 2025; Shao et al., 2025, *inter alia*), which argue that complex, multi-aspect objectives are better learned from structured criteria than from a single scalar judge. Our results suggest that even a coarse three-dimensional rubric is sufficient to recover meaningful gains in both faithfulness and accuracy. We view this as evidence for a broader principle: as outcome-only rewards saturate, the next axis of improvement for agentic systems may lie in spending more compute on *evaluation* — via richer, finer-grained, possibly automatically-generated rubrics — and propagating those signals back into training. Future work could focus on automatically generating query-conditioned faithfulness rubrics, scalable methods for distilling such fine-grained rewards, and extending this paradigm beyond agentic search to other long-horizon reasoning tasks.

### Limitations and Potential Risks

While our work demonstrates the benefits of incorporating faithfulness rewards, we acknowledge that there are still several limitations in our work. First, our evaluation of Information-Think Faithfulness relies on an LLM-as-a-Judge, which we subsequently use our trained reward model. Although we show this approach is effective, it is inherently subject to the biases and potential errors of the judge model. It is worth exploring

the development of more objective, non–model-based metrics for evaluating faithfulness in agentic search models. Second, our Think-Answer Faithfulness metric is based on substring exact-match (Sub-EM), which is precise but may lack recall. It can fail to recognize legitimate paraphrasing or logical inference, potentially treating a faithful answer as unfaithful. More sophisticated semantic matching techniques could provide a more robust evaluation. Lastly, our experiments were conducted on open-domain question-answering tasks. The effectiveness of our proposed rewards may vary in other domains, such as enterprise search or medical QA, where the nature of evidence and reasoning can be substantially different. Future work should explore the generalizability of these faithfulness-aware training methods to a wider range of applications.

We use public benchmarks licensed for academic usage. Our small scale human evaluation is conducted by an author of this paper, who is a trained NLP researcher. To the best of our knowledge, this paper does not incur potential risks and ethical concerns.

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

## A  Prompt Template in Search-R1

We show the original SEARCH-R1 template in Table 5.

Table 5: Prompt template in SEARCH-R1 (Jin et al., 2025b).

---

Answer the given question. You must conduct reasoning inside `<think>` and `</think>` first every time you get new information. After reasoning, if you find you lack some knowledge, you can call a search engine by `<search>` query `</search>`, and it will return the top searched results between `<information>` and `</information>`. You can search as many times as you want. If you find no further external knowledge needed, you can directly provide the answer inside `<answer>` and `</answer>` without detailed illustrations. For example, `<answer>` xxx `</answer>`.
**Question**: Question Content.

---

## B  Prompt Template for Faithfulness Evaluation

We show the prompt template used in Claude and our trained judge model in Table 6 and Table 7.

Table 6: Prompt template for Think-Search Faithfulness.

You are a helpful judge. The content between `<think></think>` is a language model's reasoning process after receiving new information. The content between `<search></search>` is the search query generated by this model after its reasoning process. Output 1 if the search query is clearly based on and follows from its reasoning process. Output 0 if the search query does not align with or follow from its reasoning process. The content is:
{input_string}.

Table 7: Prompt template for Information-Think Faithfulness.

You will be given some content containing the information from a retriever and the thinking process of a language model. The content between `<information></information>` is the retrieved information given by a retriever. The content between `<think></think>` is language model's reasoning process after seeing the retrieved information.
Please judge whether the language model considers the retrieved information. Output 1 if the reasoning process considers the retrieved information. Output 0 if the reasoning process does not consider the retrieved information. The content is:
{input_string}.
Please only output the score number.

## C  Details about PPO

The objective function with search engine using PPO is formulated as (Jin et al., 2025b):

$$
\mathcal{J}_{\mathrm{PPO}}(\theta) = \mathbb{E}_{x\sim\mathcal{D},\, y\sim\pi_{\mathrm{old}}(\cdot|x;\mathcal{R})} \left[ \frac{1}{\sum_{t=1}^{|y|} I(y_t)} \sum_{\substack{t=1 \\ |y_t|=1}}^{|y|} \right.
$$
$$
\left. \min\left( \frac{\pi_\theta(y_t \mid x, y_{<t}; \mathcal{R})}{\pi_{\mathrm{old}}(y_t \mid x, y_{<t}; \mathcal{R})} A_t, \mathrm{clip}\left( \frac{\pi_\theta(y_t \mid x, y_{<t}; \mathcal{R})}{\pi_{\mathrm{old}}(y_t \mid x, y_{<t}; \mathcal{R})}, 1-\epsilon, 1+\epsilon \right) A_t \right) \right],
$$

where $\pi_\theta$ is the current policy model and $\pi_{old}$ the older one. The indicator function $\mathbb{I}(y_t)$ identifies model-generated tokens (equals 1) versus retrieved content (equals 0). The clipping parameter $\epsilon$ constrains policy updates to ensure stable optimization (Shao et al., 2024). The advantage values $A_t$ are derived through Generalized Advantage Estimation (GAE, Schulman et al., 2015). In our experiments, the reward $\mathcal{R}$ is a combination of exact match $\mathcal{R}_{EM}$ and faithfulness scores $\mathcal{R}_{faithfulness}$.

## D  Details of Datasets

We show dataset statistics and their corresponding licenses in Table 8. We use the datasets processed by FlashRAG (Jin et al., 2025c).

## E  Baselines

**Original Search-R1**   The SEARCH-R1 framework trains agentic search LLMs using reinforcement learning. It uses both PPO (Schulman et al., 2017) and GRPO (Shao et al., 2024). The training data for SEARCH-R1 consists of a mixture of Natural Questions (NQ) (Kwiatkowski et al., 2019) and HotpotQA (Yang et al., 2018).

**Other baselines.**   Other baselines can be classified into three main categories: (1) Prompt-based methods: This type of methods do not have access to external knowledge source. We include both direct inference and Chain-of-Thought prompting (Wei et al., 2022). (2) Retrieval-based method: These methods are also

Table 8: Statistics and Licenses of used QA Datasets . † denotes in-domain datasets and * denotes out-of-domain datasets.

| Dataset | # Train | # Val | # Test | Corpus | Task | License |
|---|---|---|---|---|---|---|
| NQ[†] Kwiatkowski et al. (2019) | 79,168 | 8,757 | 3,610 | Wikipedia | QA | Apache 2.0 |
| TriviaQA[*] Joshi et al. (2017) | 78,785 | 8,837 | 11,313 | Wikipedia & Web | QA | Apache 2.0 |
| PopQA[*] Mallen et al. (2023) | – | – | 14,267 | Wikipedia | QA | MIT |
| HotpotQA[†] Yang et al. (2018) | 90,447 | 7,405 | – | Wikipedia | Multi-hop QA | CC BY-SA 4.0 |
| 2WikiMultihopQA[*] Ho et al. (2020) | 15,920 | 1,986 | 1,996 | Wikipedia | Multi-hop QA | Apache 2.0 |
| MuSiQue[*] Trivedi et al. (2022) | 19,938 | 2,417 | – | Wikipedia | Multi-hop QA | CC BY 4.0 |
| Bamboogle[*] Press et al. (2023) | – | – | 125 | Web | Multi-hop QA | MIT |

Table 9: Preliminary performance of different reward models against Claude-3.7-Sonnet, in the Info-Think faithfulness dimension. **Bold** denotes the best result and underline denotes second best.

| Method | Consistency Ratio↑ | Kappa Coefficient↑ |
|---|---|---|
| Qwen2.5-7B-Inst | 0.718 | 0.436 |
| + Lora Fine-tuned | 0.847 | 0.694 |
| Qwen2.5-14B-Inst | 0.784 | 0.568 |
| + Lora Fine-tuned | 0.875 | 0.75 |
| Qwen3-4B | 0.673 | 0.346 |
| + LoRA Fine-tuned | 0.861 | 0.721 |
| Qwen3-8B | 0.668 | 0.355 |
| + Lora Fine-tuned | 0.852 | 0.704 |
| Qwen3-14B | 0.818 | 0.636 |
| + Lora Fine-tuned | 0.876 | 0.751 |
| Claude-3.7-Sonnet | **0.951** | **0.915** |

training-free but can use external knowledge. We include Search-o1 (Li et al., 2025), IRCoT (Trivedi et al., 2023) and RAG (Lewis et al., 2020) as our baseline methods. (3) Training-based methods: In line with SEARCH-R1 (Jin et al., 2025b), we examine these methods: SFT, RL w/o a search engine (Guo et al., 2025), and rejection sampling leveraging a search engine (Ahn et al., 2024). All baseline results are taken from Jin et al. (2025b).

# F  Reward Model Scaling

## F.1  Preliminary Studies

In our preliminary studies, we collect 8K instances from Claude-3.7-Sonnet and split into 6K train set and 2K evaluation set, using reasoning traces from SEARCH-R1-Qwen2.5-Base-7B. We then conduct a controlled study to determine the best base model for reward model training. Table 9 reports the results compared to Claude-3.7-Sonnet. We notice that Qwen2.5-14B-Inst achieves comparable performance to Qwen3-14B (Yang et al., 2025). Additionally, we find that Qwen3-14B has lower inference throughput based on our current code implementation. Therefore, we opt for Qwen2.5-14B-Inst as our final Info-Think reward model.

## F.2  Scaling Reward Model Training Data

Table 10 reports the performance of the trained reward model in Info-Think faithfulness dimension with more data. We note that the claude model has different output across different trials of same prompts, even if we set temperature to 0, likely due to randomness in the model itself and the inference process (Chann, 2023; Anthropic, 2025; He & Lab, 2025). Table 11 are the human evaluation results. An author of this paper annotated 100 samples and compares against Claude-3.7-Sonnet, Claude-4.5-Sonnet and our trained reward model. Both LLM judges and our distilled reward model show high agreement with human annotations, and the distilled RM achieves the strongest consistency (0.910) and Cohen's $\kappa$ (0.819).

Table 10: Performance of different reward models on Info-think Faithfulness against Claude-Sonnet-3.7 (3000 samples). **Bold** denotes the best result and underline denotes second best.

| Model | Consistency Ratio↑ | Kappa Coefficient↑ |
|---|---|---|
| Qwen2.5-7B-Inst | 0.687 | 0.374 |
| + Lora Fine-tuned | 0.874 | 0.747 |
| Qwen2.5-14B-Inst | 0.753 | 0.506 |
| + Lora Fine-tuned | 0.899 | 0.797 |
| Claude-3.7-Sonnet | **0.920** | **0.840** |
| Claude-4.5-Sonnet | 0.915 | 0.829 |

Table 11: Human Evaluation Results (100 Samples)

| Model | Consistent Ratio↑ | Kappa Coefficient↑ |
|---|---|---|
| Claude-3.7-Sonnet | 0.850 | 0.699 |
| Claude-4.5-Sonnet | 0.870 | 0.739 |
| Fine-tuned Qwen2.5-14B-Inst | **0.910** | **0.819** |

# G  Additional Results

We show the training dynamics of VERITAS Info-Think in Figure 5.

# H  Case Studies

We extract some examples in the Search-R1's trajectories as case studies for reasoning faithfulness.

Figure 6 shows an example for Information-Think unfaithfulness. The retrieved information already contains the information needed, i.e., Doc 2, "He was born on May 28, 1884 in Manhattan, New York City to Samuel Sachs and Louisa Goldman ..."; but the model thoughts deem "there is no information about Louisa Goldman's husband and his title...", which is clearly contradicting the evidence.

Figure 7 shows an example for Think-Answer unfaithfulness. The final reasoning introduces concepts (parapsychology, real-time media) that are irrelevant or unsupported, and it does not logically justify the answer "reality television." This is likely a drift or hallucinated chain of thought.

Figure 6 and Figure 7 both show the examples of the policy model's Think-Search faithfulness. The model first reason about the information it already acquires (in the case of the think block after the information block), then reason about the next query it wants to issue. This observation is aligned with our findings in Figure 1, where the model exhibits high Think-Search faithfulness as indicated by both LLM-as-a-Judge and the NLI classifier. The main reason for Think-Search unfaithfulness as indicated in our evaluation results is due to the model's lack of reasoning of the next query instead of falsified or logically incorrect reasoning, which we consider as *not unfaithful*.

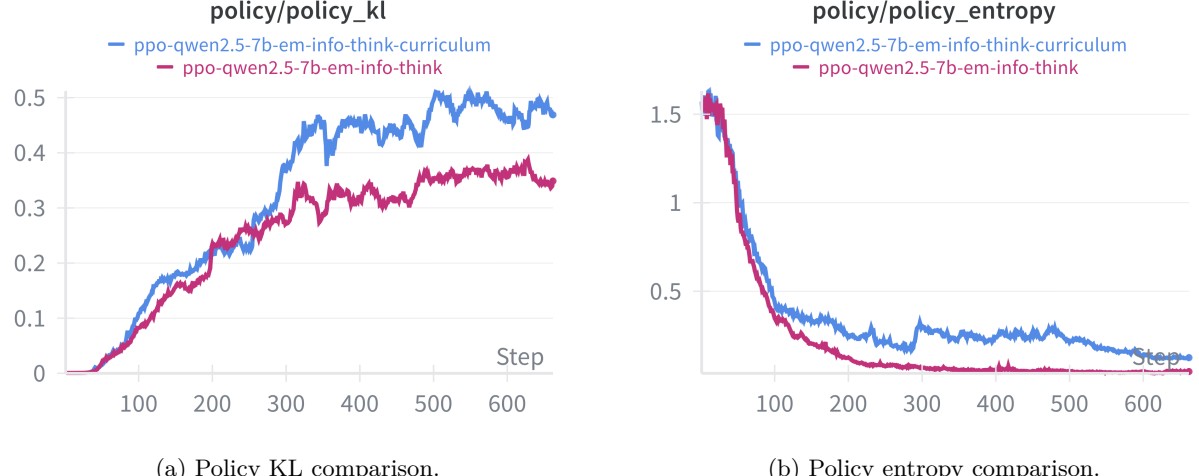

(a) Policy KL comparison.

(b) Policy entropy comparison.

Figure 5: Training dynamics of VERITAS Info-Think, with and without curriculum. Including the Info-Think faithfulness reward at the start of the training limits the policy model's exploration (lower KL in the left panel) and leads to a more deterministic policy (lower entropy in the right panel).

---

**Info-Think Unfaithfulness Example**

```
<|im_start|>system
You are a helpful assistant. First think through the reasoning internally, then provide the
    answer as instructed.
<|im_end|>

<|im_start|>user
Answer the given question. You must conduct reasoning inside <think> and </think> first every
    time you get new information. After reasoning, if you find you lack some knowledge, you
    can call a search engine by <search> query </search> and it will return the top searched
    results between <information> and </information>. You can search as many times as you
    want. If you find no further external knowledge needed, you can directly provide the
    answer inside <answer> and </answer>, without detailed illustrations. For example,
    <answer> Beijing </answer>. Question: What was Louisa Goldman's husbands title?

<think> First, I need to determine who Louisa Goldman is and find out information about her
    husband's title. </think>

<search> Louisa Goldman's husband's title </search>

<information>
Doc 1: "Walter E. Sachs"\nWalter E. Sachs Walter Edward Sachs (May 28, 1884  August 21, 1980)
    was an American banker and financier. He was born on May 28, 1884 in Manhattan, New York
    City to Samuel Sachs and Louisa Goldman of the GoldmanSachs family. He was an alumnus of
    Harvard Business School. Sachs was a partner at Goldman Sachs starting in 1928, guiding
    the company through the Great Depression. He married Mary Williamson in 1939 and divorced
    in 1960. He retired as a partner in 1959 to become a limited partner. He died at his home
    in Darien, Connecticut on August 21, 1980.
Doc 2: "Walter E. Sachs"\nWalter E. Sachs Walter Edward Sachs (May 28, 1884  August 21, 1980)
    was an American banker and financier. He was born on May 28, 1884 in Manhattan, New York
    City to Samuel Sachs and Louisa Goldman of the GoldmanSachs family. He was an alumnus of
    Harvard Business School. Sachs was a partner at Goldman Sachs starting in 1928, guiding
    the company through the Great Depression. He married Mary Williamson in 1939 and divorced
    in 1960. He retired as a partner in 1959 to become a limited partner. He died at his home
    in Darien, Connecticut on August 21, 1980.
Doc 3: "Amy Goldman Fowler"\nAmy Goldman Fowler Amy Goldman Fowler (born 1954) is an American
    billionaire heir, gardener, author, artist, philanthropist, and advocate for seed saving
    and heirloom fruits and vegetables. She is one of the foremost heirloom plant
    conservationists in the US. Goldman is described as ""perhaps the world's premier
    vegetable gardener"" by Gregory Long, President of The New York Botanical Garden. On
    April 28, 2012, Goldman married Cary Fowler at the terrace on top of the Arsenal in
    Central Park. Fowler is the daughter of Lillian (née Schuman) and Sol Goldman. She has
    three siblings: Allan H. Goldman, Diane Goldman Kemper, and\n"</information>

<think> Based on the search results, it is clear that the information is mostly about Walter
    E. Sachs, who was indeed related to the Goldman family. However, there is no information
    about Louisa Goldman's husband and his title. Let's try a different approach by searching
    for information about Louisa Goldman. </think>

<search> Louisa Goldman's husband's title </search>

...

<answer> partner </answer>
```

Figure 6: Example of Information-Think unfaithfulness in Search-R1.

---

**Think-Answer Unfaithfulness Example**

```
<|im_start|>system
You are a helpful assistant. First think through the reasoning internally, then provide the
    answer as instructed.
<|im_end|>

<|im_start|>user
Answer the given question. You must conduct reasoning inside <think> and </think> first every
    time you get new information. After reasoning, if you find you lack some knowledge, you
    can call a search engine by <search> query </search> and it will return the top searched
    results between <information> and </information>. You can search as many times as you
    want. If you find no further external knowledge needed, you can directly provide the
    answer inside <answer> and </answer>, without detailed illustrations. For example,
    <answer> Beijing </answer>. Question: Wim Schuhmacher is mostly associated with a genre
    sometimes called what ?

<think>1. We need to identify the genre associated with Wim Schuhmacher.</think>

<search> query: Who is Wim Schuhmacher and what genre is he associated with? </search>

<information>...</information>

<search> query: what genre to explain Wim Schuhmacher? </search>

<information>...</information>

<think>i found that he is a known filmmaker in the field, videos (a short video in
    particular)</think>

<search> query: It is Real Time Part. It is the same for the other genre? </search>

<information>
Doc 1: "Real-time computing"\nas ""real-time"" would be more accurately described as ""near
    real-time"". Near real-time also refers to delayed real-time transmission of voice and
    video. It allows playing video images, in approximately real-time, without having to wait
    for an entire large video file to download. Incompatible databases can export/import to
    common flat files that the other database can import/export on a scheduled basis so that
    they can sync/share common data in ""near real-time"" with each other. The distinction
    between ""near real-time"" and ""real-time"" varies, and the delay is dependent on the
    type and speed of the transmission. The delay in near real-time is
Doc 2: "Real time (media)"\nReal time (media) Real time within the media is a method where
    events are portrayed at the same rate at which the characters experience them. For
    example, if a movie told in real time is two hours long, then the plot of that movie
    covers two hours of fictional time. If a daily real-time comic strip runs for six years,
    then the characters will be six years older at the end of the strip than they were at the
    beginning. This technique can be enforced with varying levels of precision. In some
    stories, every minute of screen time is a
Doc 3: "Internet genre"\n...
</information>

<think>the right genre combination is parapsychology, he is associated specifically with a
    genre sub Real Time Part.</think>

<answer> reality television </answer>
```

Figure 7: Example of Think-Answer unfaithfulness in Search-R1.

