# OpenReview forum: "Beyond Correctness: Rewarding Faithful Reasoning in Retrieval-Augmented Generation"
_TMLR — Accepted by TMLR_

### Review · Reviewer_yaaB · 2026-02-12

**Summary Of Contributions:**

1. Identified faithfulness gaps in existing RL search agents and established a comprehensive evaluation framework.

2. Proposed VERITAS, a novel training method that integrates fine-grained faithfulness rewards into the RL process.

3. Demonstrated that this approach significantly improves both intermediate reasoning reliability and final task performance.

**Audience:**

Yes

**Audience Explanation:**

The topic is popular and important.

**Broader Impact Concerns:**

No Concern

**Claims And Evidence:**

Yes

**Claims Explanation:**

The paper's motivtion is clear and easy to follow.

**Requested Changes:**

### **Typos**

1.  **"an** information block" (Changed "a" to "an")
2.  "The cost and latency of the LLMaaJ approach" (LLM-as-Judge)

### **Questions**

1.  Could the authors elaborate on the differences between the proposed framework and the standard GRPO method used in DeepSeek-R1?
2.  I would like to ask if the LLM-as-a-Judge component is prone to hallucination.
3.  The reward function involves several hyper-parameters ($w$). How were these determined, and is the performance sensitive to these weight settings?
4.  I am also curious about the influence of the reward model's size on the final performance of the search agents.

---

### Review · Reviewer_xrLD · 2026-02-22

**Summary Of Contributions:**

This paper addresses the gap between task accuracy and reasoning faithfulness in RL-trained search agents. The authors propose (1) a three-dimensional faithfulness evaluation framework and (2) VERITAS, a training framework that incorporates faithfulness metrics as process-based rewards into PPO-based RL. Experiments on seven QA benchmarks show VERITAS-R1 improves both faithfulness and task accuracy over Search-R1.

---


Strengths

- The decomposition of faithfulness into three trajectory-level dimensions is intuitive, well-grounded in the agentic search structure, and practically useful.
- The systematic audit of existing models (Search-R1, ReSearch, DeSA) reveals a previously under-examined failure mode: outcome-correct but process-unfaithful reasoning. The finding that format rewards degrade faithfulness is a particularly valuable insight for the community.
- The empirical result that process supervision improves both faithfulness and EM is the paper's strongest finding and challenges the implicit assumption that these objectives trade off.


Weaknesses

1. Faithfulness metrics are largely LLM-judge-dependent. Info-Think faithfulness is evaluated by an LLM judge, then distilled into an RM that is itself used to train the policy. The paper never rigorously establishes what "faithful" means beyond "what Claude-3.7-Sonnet labels as faithful." The 50-sample human evaluation conducted by one of the authors (Table 10) is far too small to validate this entire pipeline. A larger-scale, multi-annotator human study with inter-annotator agreement is needed.
2. The authors acknowledge that R_think-answer "does not reliably improve its corresponding metric and can sometimes be detrimental", yet the final VERITAS configuration still includes it (EM+Info-Think+Think-Ans). The rationale for retaining an unreliable reward signal is unclear, and the ablation in Table 3 shows the combined variant rarely outperforms Info-Think alone.
3. All experiments use a single base model (Qwen2.5-7B-Base), a single RL algorithm (PPO), and short-form open-domain QA. Generalization to larger models, other model family, GRPO-like algo, is unexamined. The generalizability of the framework is thus unsupported.


Minor
1. Several typos and grammatical errors (e.g., "consistenting" in §2; "futher enhance" in §5.2; incomplete sentence in §2 "RL to d-based reasoning").

**Audience:**

Yes

**Audience Explanation:**

The paper is at the intersection of RL for LLMs, RAG, and reasoning faithfulness, which are three highly active areas. The finding that outcome-correct agents produce unfaithful reasoning, and that process supervision helps both faithfulness and accuracy, is practically relevant to anyone building or deploying agentic search systems. The faithfulness evaluation framework fills a gap the community currently lacks.

**Claims And Evidence:**

No

**Claims Explanation:**

The evidence is directionally consistent but falls short of rigor on several key claims.

The faithfulness evaluation framework is the paper's foundation, but relies on LLM-judge labels. Info-Think has no non-LLM-based metric, making the entire measurement framework dependent on trusting Claude's judgments.

As for the experiment results, the faithfulness improvement from VERITAS is visually clear in Figure 3 and Table 3, but suffers from circularity: the evaluation metric is the same used to train the policy. The accuracy improvement is the weakest. Margins are small (0.431→0.447 avg EM), no significance tests or confidence intervals are reported, and the authors themselves acknowledge vLLM randomness causes cross-table inconsistencies. This needed multi-seed runs with error bars.

On methods, the curriculum learning design is stated as hypothesis but the ablation is not shown.

**Requested Changes:**

- Report faithfulness results using an evaluation metric independent of the training RM — either a held-out LLM judge not involved in distillation, or a substantially larger human evaluation.
- Statistical rigor for EM claims. Provide multi-seed runs with standard deviations or confidence intervals. The claimed accuracy gains (0.431→0.447) are small enough to fall within noise acknowledged by the authors themselves.
- Scale human evaluation. 50 samples by a single author-annotator is insufficient to support the entire faithfulness measurement pipeline.
- Ablate curriculum learning. Show the no-curriculum baseline explicitly. This is a key design choice without experiment justification.
- More case studies to help the reader better understand the three faliure modes.

---

### Review · Reviewer_HA7N · 2026-03-11

**Summary Of Contributions:**

This paper addresses the critical issue of unfaithful intermediate reasoning in Reinforcement Learning based RAG systems, which are often referred to as agentic search. The authors correctly point out that current models, which are optimized primarily for final answer correctness, tend to produce reasoning traces that misalign with the retrieved evidence or the final output.

The paper identifies a crucial limitation in current RL-based RAG models and provides a structured approach to measure it.

The proposed evaluation framework offers a clear, theoretical grounding by breaking down agentic search trajectories into verifiable dimensions.

The authors introduce a practical and effective training paradigm, which successfully integrates fine-grained, process-based faithfulness rewards into the RL training loop without sacrificing scalability.

**Audience:**

Yes

**Audience Explanation:**

TMLR audience would be interested in the findings of this paper

**Claims And Evidence:**

Yes

**Claims Explanation:**

Empirical results validate the central hypothesis, demonstrating that the VERITAS framework not only enhances the reasoning faithfulness of the models but also synergistically improves overall task performance across diverse QA datasets.

**Requested Changes:**

The reliance on an LLM-as-a-Judge to generate the training data for the distilled reward model introduces potential biases. The paper would benefit from a slightly expanded discussion on the limitations of model-based evaluation and potential steps toward more objective metrics.

The authors acknowledge that the Think-Answer faithfulness metric relies on a substring exact-match approach, which may lack recall and unfairly penalize legitimate paraphrasing. Adding a brief discussion on how future work might integrate more sophisticated semantic matching could improve the methodology section.

Clarify the justification for setting the Think-Search reward weight to zero during the VERITAS training phase, ensuring readers fully understand why this specific dimension did not require active optimization despite being part of the evaluation framework.

---

### Decision · Action_Editor_sybJ · 2026-04-30

**Recommendation:** Accept with minor revision

**Additional Comments:**

The evaluation relies too heavily on LLM-judge labels without an independent, non-LLM-based metric or a large-scale human study.

The faithfulness improvements suffer from circularity because the evaluation metric is the same one used to train the policy.

The accuracy improvement claims lack statistical rigor, pointing out that the margins are small and missing multi-seed runs with error bars.

The curriculum learning design lacks an explicit ablation study without the curriculum baseline.

**Audience:**

Yes

**Audience Explanation:**

Researchers and practitioners within the TMLR audience who are focused on NLP and LLMs would likely find the findings highly relevant.

**Claims And Evidence:**

Yes

**Claims Explanation:**

The empirical results validate the central hypothesis, demonstrating that the VERITAS framework successfully enhances both the reasoning faithfulness of the models and overall task performance across diverse QA datasets. The paper's motivation is clear and easy to follow.

---

> ### Author Response · Authors · 2026-05-26
> **Response to AE and Updates w.r.t. Camera Ready Version**
>
> We thank the Action Editor and the Reviewers for comprehensive feedback over the manuscript. We have updated the manuscript as the camera ready version. Please refer to #Changes Since Last Submission for detailed changes.